# New Indications of Biological Drugs in Allergic and Immunological Disorders: Beyond Asthma, Urticaria, and Atopic Dermatitis

**DOI:** 10.3390/biomedicines11020236

**Published:** 2023-01-17

**Authors:** Daniele Russo, Paola Di Filippo, Sabrina Di Pillo, Francesco Chiarelli, Marina Attanasi

**Affiliations:** Department of Pediatrics, University of Chieti, 66100 Chieti, Italy

**Keywords:** monoclonal antibodies, chronic rhinosinusitis with nasal polyps, eosinophilic esophagitis, food allergy, oral immunotherapy

## Abstract

Asthma, chronic urticaria, and atopic dermatitis are some of the most numerous allergic diseases affecting children. Recent advances in the understanding of their specific intracellular molecular pathways have led to the approval of monoclonal antibodies targeting definite inflammatory molecules in order to control symptoms and improve quality of life. Less is known about other allergic and immunologic disorders such as rhinosinusitis with nasal polyps, eosinophilic esophagitis, anaphylaxis, and food allergy undergoing allergen immunotherapy. The increasing evidence of the molecular mechanisms underlying their pathogeneses made it possible to find in children new indications for known biological drugs, such as omalizumab and dupilumab, and to develop other ones even more specific. Promising results were recently obtained, although few are currently approved in the pediatric population. In this review, we aim to provide the latest evidence about the role, safety, and efficacy of biologic agents to treat allergic and immunologic diseases in children.

## 1. Background: Where We Start From

Recent insights in the pathophysiology of allergic disorders allowed novel therapeutic strategies for the treatment of different allergic diseases in the pediatric population to be identified. This favorable trend aims to positively change the natural history of atopic children and to improve their quality of life. Indeed, the increasing characterization of the exact molecular pathways of asthma, atopic dermatitis (AD), and chronic spontaneous urticaria (CSU) permitted the development of several biologic drugs targeting precise different molecules enrolled in the inflammatory allergic cascade. As a matter of fact, in our previous narrative review [1], we deeply investigated the wide use of monoclonal antibodies (mAbs) in severe asthma, according to step 5 of the Global Initiative for Asthma. In particular, we discussed the therapeutic use of omalizumab, IL-5 blockers, and anti-IL-4Rα mAb.

Omalizumab is the anti-IgE mAb currently indicated for T2 high allergic asthma from six years of age, IL-5 blockers, such as mepolizumab from six years of age and benralizumab from twelve years of age, and anti-IL-4Rα mAb as dupilumab from six years of age are instead approved for T2 high eosinophilic asthma. Recent evidence also demonstrated the potential role and effectiveness of new biological drugs directed against specific type 2 alarmin cytokines (i.e., thymic stromal lymphopoietin [TSLP], IL-25, and IL-33) in severe asthma [1,2]. These molecules are released from epidermal, epithelial, and stromal cells after biological and chemical insults from the surrounding environment—the exposome, as an attempt of the body to maintain tissue homeostasis [3]. These cytokines activate dendritic cells (DCs) to stimulate T helper 2 (TH2) cells and activate group 2 innate lymphoid cells (ILC2s) to shift the immune system to type 2 responses. Therefore, a complex interaction occurs between DCs, ILC2, and TH2, with the subsequent massive release of allergic cytokines, such as IL-4, IL-5, IL-9, and IL-13 which participate in the inflammatory response [4]. In this perspective, new clinical trials recently investigated new biologics as Tezepelumab, an anti-TSLP, and its ability to reduce the annualized rate of asthma exacerbations and to greatly improve lung function compared to placebo [5]. In addition, Kelsen et al. [6] carried out a double-blind, placebo-controlled, dose-ranging study involving 502 adults with severe asthma who were administered different doses of astegolimab. Astegolimab is a novel biologic drug blocking IL-33R (also called ST2) and its intracellular pathway is implicated in the onset of eosinophilic asthma. Through its mechanism, it reduced the annualized asthma exacerbation rate in a broad population of patients [6]. Another phase two trial with Itepekimab, a mAb targeting IL-33, showed similar results with improvement of lung function in patients with moderate-to-severe asthma (NCT03387852) [7]. However, these promising results in adults need to be evaluated in children with moderate–severe asthma, in order to ensure an increasing availability of biologics in the so-called precision medicine of severe asthma.

Concerning AD, dupilumab has been widely shown to play a key role in the pathogenesis, as it specifically targets the heterodimeric receptor IL-4Rα and thereby blocks IL-4 and IL-13, preventing intracellular inflammatory signal transduction [8]. Recently, dupilumab has been approved for ages six and up. However, Paller et al. [9] lately conducted the first randomized, double-blind, placebo-controlled, phase three trial involving one hundred and sixty-two patients aged six months to six years, with moderate-to-severe AD. These patients randomly received dupilumab (*n* = 83) or placebo (*n* = 79) as an adjunctive therapy to local steroids. It was seen that a large amount of patients treated with dupilumab recorded at week 16 a significant improvement of AD signs and symptoms, in terms of the Investigator Global Assessment (IGA) 0–1 (28% vs. 4%, 95% CI 13–34; *p* < 0.0001) and Eczema Area and Severity Index (EASI–75) (53% vs. 11%, 95% CI 29–55; *p* < 0.0001). In addition, no significant differences about the number of adverse events were found between the two groups (64% with dupilumab vs. 74% of placebo patients), whereas a higher incidence of conjunctivitis was recorded in the dupilumab group compared to the placebo group (5% vs. 0%).

Currently, despite an increasing interest towards the key role of IgE and eosinophils in the pathogenetic pathways of AD, omalizumab and mepolizumab have not been approved yet. Moreover, novel pivotal mAb are currently under investigation in AD, such as nemolizumab, lebrikizumab, etokimab, fezakinumab, and tralokinumab, although more safety, efficacy, and longer studies are necessary to better understand their possible use in AD in children. [8].

New insights in the pathogenesis of CSU allowed several biologic therapies acting in its inflammatory pathways to be developed. Recently, a new mAb called lirentelimab has been studied, as it targets sialic-acid-binding Ig-like lectin 8 (SIGLEC8), reducing the number of eosinophils and interfering with the action of mast cells. In this way, lirentelimab may play an important role in patients with antihistamine-refractory chronic urticaria [10].

Although several trials of biologic drugs are ongoing, omalizumab is nowadays the only mAb approved in children with severe and treatment-refractory CSU. Its efficacy and safety were widely demonstrated, but worldwide availability, high costs, and a not yet clear duration of treatment still represent a major limitation to its use. Ligelizumab, a new humanized mAb against IgE, may represent a promising option, but further studies are needed to better define its efficacy and safety in the pediatric population [8].

However, the identification of the ideal drug, as well as the optimal treatment duration and the high manufacturing costs, are still a matter of debate mostly due to the wide variability of treatment response.

The potential use of biologics in the context of allergic diseases beyond AD, asthma, and immunologic diseases is even less well known. In the last years, several trials investigated their potential role to target precise extra- or intracellular molecules, such as the IgE molecule or type 2 cytokines, involved in the pathogenetic immune response. Our review focuses primarily on biologics in other allergic or immunologic diseases, such as chronic rhinosinusitis with nasal polyps (CRSwNP), eosinophilic esophagitis (EoE), anaphylaxis, food allergy, and allergen immunotherapy (AIT). We also discuss how these findings contributed to our increasing understanding of the pathophysiology of allergic and immunologic disorders in children.

## 2. Methods and Materials

In this narrative review, we searched for articles on PubMed using the keywords “chronic rhinosinusitis with nasal polyps”, “eosinophilic esophagitis”, “anaphylaxis”, “food allergy”, “children”, “adolescents”, and “biologic therapies”, preferring articles published in the last ten years. Further studies were obtained through the references of some papers, in particular, meta-analyses with solid evidence published even before the time interval considered. Articles were selected according to their abstract, using eligibility criteria. The inclusion criteria were being in the English language, pediatric study population (age range 0–18 years old), and type of study: narrative and systematic reviews, retrospective analysis, and randomized control trials. Case reports, expert opinions, and manuscripts published in a language other than English were excluded. The final reference list was developed on the basis of originality and relevance to the broader scope of this review.

## 3. Chronic Rhinosinusitis with Nasal Polyps

CRSwNP is defined as a condition of chronic inflammation of the mucous membrane of the nose and paranasal sinuses [11]. CRSwNP causes symptoms such as nasal obstruction and congestion, facial pressure, and anosmia which could severely impact on the child’s sleep and quality of life.

In children, this chronic inflammation may result in progressive bilateral growth of polyps, which can be identified with direct or endoscopic examination in the middle meatus of the nasal cavity.

Little is known yet on the etiopathogenesis of sinusitis, since it could be derived from an abnormal immune response to an external stimulus through the nasal airway, such as irritants, commensal and microorganisms (fungi and *Staphylococcus aureus*), and inhalant allergens. Other causes of chronic inflammation could be derived from a mechanical blockage of the physiologic sinus airway or altered protective functions of the mucociliary function and barrier [11,12].

Oykhman et al. [13] carried out a network meta-analysis of all available mAbs for CRSwNP, including 29 randomized controlled trials and 3461 adult patients. The authors showed an improvement of health-related quality of life with dupilumab, omalizumab, mepolizumab, and benralizumab compared with placebo. In addition, they found an important reduction of the risk of rescue nasal polyp surgery thanks to the use of biologic agents. Dupilumab was found to be the most effective among all the biologics considered [13].

A systematic review of 12 randomized controlled trials involving 1527 participants has recently investigated the effects on quality of life of dupilumab, mepolizumab, and omalizumab for adult patients with CRSwNP [12]. In particular, the authors showed that dupilumab improved disease-specific health-related quality of life and contextually showed a reduction in disease severity compared with placebo. Concerning mepolizumab and omalizumab, the data suggested a low improvement of quality of life, with no clear effects on disease severity [12].

Recently, Bachert et al. [14] conducted a clinical trial (NCT03401229) which studied the role of benralizumab in patients with CRSwNP. The authors enrolled 413 symptomatic adults with severe CRSwNP who were not responsive to conventional therapies of intranasal corticosteroids or systemic corticosteroid use and/or surgery for nasal polyps (NP). After 1:1 randomization to treatment with benralizumab 30 mg or placebo every 4 weeks at the beginning and then every 8 weeks, specific endpoints in NP score (NPS) at 40 weeks and patient-reported symptoms were evaluated. Benralizumab showed a significant efficacy in improvement of NPS and nasal obstruction score compared to placebo. Authors showed that no statistically significant differences between treatment groups occurred in terms of changes of Sino-Nasal Outcome Test-22 (SNOT-22) score at week 40 or time to first nasal polyp surgery. Therefore, this study concluded that benralizumab, when added to standard-of-care therapy in patients with CRSwNP, reduced NPS, nasal blockage, and difficulty with sense of smell compared to placebo but did not significantly change time to first polyp surgery.

Similar results in terms of reduction of nasal polyp size and improvement of nasal symptoms were provided in another recent phase three study (the SYNAPSE study [NCT03085797]) investigating the role of mepolizumab in CRSwNP [15].

Efficacy and safety of omalizumab for nasal polyposis were also investigated. In an open-label extension study conducted by Gevaert et al. [16], on-treatment patients continued to benefit from omalizumab treatment despite discontinuation at 52 weeks. Authors recorded a gradual worsening of clinical assessment over the 24-week follow-up but still improved compared to pretreatment levels for both groups. In this way, this study made it clear that omalizumab treatment enjoyed a sustained efficacy and safety profile for up to 1 year in patients with CRSwNP and, at the same time, showed an insufficient response to nasal corticosteroids [16].

To date, finding specific molecular markers characterizing precise molecular endotypes is the greatest challenge in children with CRSwNP. However, further studies specifically conducted in the pediatric population are needed. In this way, it would be easier to choose the most appropriate biologic drug to effectively control symptoms of this wide-spread disease and its perspective and risks of surgery. According to this, new indications for the choice of the most appropriate biologic drug in CRSwNP were moved by the European Forum for Research and Education in Allergy and Airway Diseases expert board meeting [17].

## 4. EoE

Eosinophilic esophagitis is a chronic immune-mediated inflammatory disease of the esophagus affecting both adults and children all over the world, with clinical manifestations depending on the age onset [18]. EoE is characterized by several symptoms, such as vomiting, dysphagia, or feeding difficulties, in a patient with an esophageal biopsy with at least 15 eosinophils per high-power field (hpf). Diagnosis is made by excluding all other common diseases impacting on esophageal disfunction, such as gastroesophageal reflux disease or achalasia [18].

The pathogenesis is still unclear, although both genetic and environmental factors, such as exposure to antibiotics early in life, are implicated in EoE. Indeed, the genetic predisposition leads to higher esophageal tissue permeability and the penetration of harmless antigens could generate an abnormal T-helper 2 (Th2)-immune response [19]. Therefore, several cytokines such as IL-4, IL-5, and IL-13 are released, with the subsequent recruitment of inflammatory cells involving Th2-cells, active B cells, and eosinophils [20,21,22,23,24,25]. These mechanisms lead to an upregulation of periostin and growth factor (TGF)-β, increasing the adhesion of eosinophils to fibronectin and directly inducing the expression of profibrotic genes, such as fibronectin and collagen I, resulting in esophageal epithelial fibrostenosis [26].

First-line treatment of EoE includes proton pump inhibitors, local steroid preparations, such as fluticasone and budesonide, dietary therapy with amino acid formula or food exclusion, and endoscopic dilation. Beyond conventional therapies, many novel biologic agents are also under clinical investigation.

Hirano et al. [27] conducted a multicenter clinical trial involving EoE patients treated with placebo or dupilumab weekly for 12 weeks [27]. A total of 83 percent of patients treated with dupilumab recorded an endoscopic number of eosinophils of less than 15 per hpf at week 12 compared with 0% receiving placebo (*p* < 0.0001). More than one third of patients receiving dupilumab recorded symptomatic improvement by week 10 compared with 13% receiving placebo (*p* = 0.049), with significantly better esophageal distensibility. In this study, the safety profile of dupilumab was confirmed, as local skin manifestations were the most common adverse reactions (one third in the dupilumab group vs. 8% in the placebo one) followed by inflammation of the upper airways (17% in the dupilumab group vs. 4% in the placebo one).

In May 2022, dupilumab was the first biologic agent approved by the U.S. Food and Drug Administration (FDA) to treat EoE in adults and pediatric patients 12 years and older weighing at least 40 kg. Such approval could be possible thanks to a randomized, double-blind, parallel-group, multicenter, placebo-controlled trial, that included two 24-week treatment periods (Part A and Part B) in separate groups of 321 participants older than 12 years old [28]. Enrolled patients received either placebo or 300 milligrams of dupilumab every week. Authors showed that in a large number of treated patients, dupilumab permitted the achievement of complete histological remission, as well as a significant reduction of symptoms, in terms of difficulty swallowing and pain, and therefore quality of life. Among the side effects recorded in this study, mild upper airways infections and joint pain were the most described [28].

Currently, a new phase three trial assessing the pivotal role of dupilumab in children aged 1 to 11 years with EoE is ongoing, even though it has already showed preliminary benefits in terms of histological disease remission at 16 weeks [29].

Other biologics are currently under clinical investigation. A phase two, multicenter, randomized, double-blind, placebo-controlled parallel-group clinical trial (NCT02098473) investigated the role of cendakimab (anti-IL-13 mAb RPC4046) in adults with EoE [30]. The results showed no clinical differences between two groups in endoscopic and histological scores, although a reduction of epithelial–mesenchymal transition markers and therefore mean eosinophil count was recorded. These results showed that cendakimab IL-13 inhibition could reduce chronic inflammation in EoE and subsequent degenerative wall stiffness of the esophagus [30,31]. In conclusion, dupilumab is the only biologic drug approved in children with EoE from 12 years of age and no biologic drug for children with EoE under 12 years of age was yet approved.

## 5. Anaphylaxis

The World Allergy Organization defined anaphylaxis as “a serious systemic hypersensitivity reaction that is usually rapid in onset and may cause death. Severe anaphylaxis is characterized by potentially life-threatening compromise in airway, breathing and/or the circulation, and may occur without typical skin features or circulatory shock being present” [32]. Pathogenetic mechanisms differ between IgE-mediated anaphylaxis (the most frequent in children), and non-IgE-mediated anaphylaxis, which may be immunologic or non-immunologic. Idiopathic anaphylaxis (IA) is instead defined when no evident causes can be identified, becoming therefore a diagnosis of exclusion.

The management of anaphylaxis involves both preventive measures and emergency therapies, with intramuscular epinephrine (adrenaline) that continues to be the first-line treatment. Currently, no prophylactic therapies are approved to prevent or reduce the occurrence of further episodes in recurrent anaphylaxis. To date, evidence suggests new possible implications of biologic agents targeting specific molecules involved in anaphylactic pathogenesis.

Indeed, Carter et al. [33] recently conducted a double-blind, placebo-controlled trial involving 82 patients (aged 13–70 years) with a diagnosis of recurrent IA, who were treated with omalizumab. Notwithstanding, the authors did not find a significant difference between the omalizumab and placebo group in terms of the number of anaphylactic events in the 6 months after baseline. However, a positive trend for efficacy was seen in the treatment group, particularly after 60 days, highlighting the promising role in the future of omalizumab in IA.

Recent evidence suggested that patients with IA, who were not controlled with the continuous use of oral corticosteroids and/or antihistamines, recorded no anaphylaxis episodes after treatment with omalizumab (300–375 mg every 2–4 weeks) for 10–14 months [34].

Other causes of IA could derive from immunological diseases, such as mastocytosis and mast cell (MC) activation disorders [35]. Patients with systemic mastocytosis (SM) may suffer from MC mediator-related symptoms poorly controlled by common therapy. Broesby-Olsen et al. [36] recently conducted an observational cohort study involving 14 adult SM patients who received omalizumab for a time ranging 1–73 months. The authors showed an efficacy profile treatment for recurrent anaphylaxis and skin symptoms, less for the other organs involved. Omalizumab is a conventional treatment in other MC-driven diseases, but little is known in SM. Notwithstanding, controlled studies in pediatric populations are needed to confirm these findings. Moreover, omalizumab seems to reduce anaphylaxis in patients with hereditary alpha-tryptasemia who experienced severe symptoms of MC activation [37].

To the best of our knowledge, although the current management of anaphylaxis is well established, different biologic drugs have been studied as potential preventive treatments. New insights in the underlying mechanisms and markers of anaphylaxis will permit the integration of new therapeutic approaches, such as biologic agents in the contest of precision medicine. Further well-performed controlled studies in children are needed to find accurate biologic agents able to prevent and reduce anaphylactic episodes and improve quality of life in children.

## 6. Food Allergy and Allergen Immunotherapy

Food allergy is increasing in the pediatric population in terms of gravity and incidence, which ranges from 1% to 10% in IgE-mediated forms in infants and preschool-aged children, from 1% to 5% in school age, and about 4.5% in adults [38,39,40,41]. The non-IgE-mediated forms, which typically cause gastrointestinal and skin symptoms, occur later than those affecting children with reactions 24–72 h after ingesting the causative food allergen [42].

Desensitizing allergen immunotherapy is one of the main approaches modifying the course of allergic diseases. The continuous allergenic stimulation of basophils and MCs through the administration of higher doses of food allergen in oral immunotherapy (OIT) reduce cell degranulation and release of preformed mediators. Such effect prevents the development of the allergic reaction to unintentional contact with the allergen and the consequent onset of multiorgan symptoms [43]. However, this treatment is not free from risks and has some limitations. Therefore, precision medicine can be a useful adjuvant tool to oral food immunotherapy. Currently, several biological drugs are being studied in the field of food allergies.

In the literature, omalizumab has been proved to reduce the risk of anaphylactic events during OIT, even though the right dosage approach is still not clear. Wood et al. [44] conducted a pivotal double-blind, placebo-controlled trial involving 57 patients ranging from 7 to 32 years old with IgE-mediated cow’s milk allergy (CMA). The primary goal of this study was to evaluate if adjunctive treatment with omalizumab proved to increase the safety of patients during OIT, even in terms of higher efficacy. Subjects were randomized 1:1 to receive omalizumab or placebo, both in combination with milk OIT. Significantly fewer milk OIT doses allowed symptom control in the omalizumab group to be gained (median 198.0 vs. 225.0; *p* = 0.008), with a subsequent shorter escalation phase (median 25.9 vs. 30.0 weeks; *p* = 0.01). However, no significant difference in terms of prolonged benefits in the treated group was found after the discontinuation of treatment. Notwithstanding, the authors showed a marked reduction of adverse events during OIT escalation in omalizumab-treated subjects for percentages of doses per subject provoking symptoms (2.1% vs. 16.1%, *p* = 0.0005) and dose-related reactions requiring treatment (0.0% vs. 3.8%, *p* = 0.0008). This study showed that omalizumab, in combination to OIT, seemed to provide a better response in terms of safety in children with CMA during milk OIT, but not in terms of efficacy [44]. Higher safety and tolerability were obtained in two further studies involving children affected by peanut and multiple-food allergens who underwent OIT in combination with omalizumab [45,46]. As a matter of fact, a double-blind, placebo-controlled clinical trial involving 36 high-risk subjects with peanut allergy between 7 and 25 years of age called the PRROTECT study (Peanut Reactivity Reduced by Oral Tolerance in an Anti-IgE Clinical Trial) was conducted. This trial investigated the effect of pretreatment of omalizumab on patients who undergo OIT, showing a rapid acquisition of desensitization and a faster tolerance to higher doses of peanut of about 10 times as compared to those treated with only placebo [47]. In addition, a multicenter, randomized, double-blind, placebo-controlled study called OutMATCHT (Omalizumab as Monotherapy and as Adjunct Therapy to Multi-Allergen OIT in Food Allergic Children and Adults) is currently ongoing [48]. This study involves 225 participants from 1 to 56 years of age with multiple food allergies and aims to test omalizumab monotherapy and as an add-on treatment to multi-food OIT. The results will provide new data about the role of omalizumab as an adjunctive treatment to OIT and as monotherapy in food allergy, and thus in the prevention of anaphylaxis caused by food [48].

Regarding other biologics, two studies are currently underway to evaluate the use of dupilumab in combination with OIT or as a monotherapy in children with peanut allergy [49,50]. The hope is to have as many tools as possible in the context of precision medicine that can reduce or prevent the risk of severe reactions in allergic children.

Concluding, no biologic drug is yet approved in food allergy management, but omalizumab could be useful to reduce the risk of anaphylactic events during OIT in children.

## 7. Conclusions

To date, many studies have been developed and others are currently underway evaluating the role, efficacy, and new indications of increasingly innovative and precise biological drugs in children, as summarized in Table 1. The challenge in the context of precision medicine of allergic and immunological diseases is to better understand the molecular mechanisms that underlie these widespread disorders. The identification of precise molecular targets could make it increasingly easier to use ad hoc biological drugs to counteract the pathogenic cascade and to change the course of allergic and immunological diseases from the first years of the child’s life. In addition, the recent development of many studies in children could permit the current limits to be overcome in terms of very high costs, adverse reactions (albeit rare), and different efficacy of biologics in different groups of children worldwide.

Recently, several biological drugs were approved in children with asthma, atopic dermatitis, and chronic urticaria. However, the use of these drugs in allergic and immunological diseases other than those aforementioned has been less investigated and therefore represents a field constantly growing.

Dupilumab seems to be the most suitable biological drug for CRSwNP, although also benralizumab and mepolizumab seem to be effective in clinical improvement. To date, dupilumab is approved by the FDA also to treat EoE in children from 12 years of age and a pivotal phase three trial in children aged 1 to 11 years is ongoing. Furthermore, other biologics for EoE in children are currently under investigation. No biologic drug is yet approved in anaphylaxis and food allergy management, but omalizumab seems to be useful to reduce the number of anaphylaxis episodes and the risk of anaphylactic events during OIT.

However, further studies specifically conducted in the children are needed in order to choose the most appropriate biologic drug in these diseases.

## Figures and Tables

**Table 1 biomedicines-11-00236-t001:** Main biologics currently approved and on study in allergic and immunologic diseases in children. CRSwNP = chronic rhinosinusitis with nasal polyps; EoE = eosinophilic esophagitis; IL = interleukin.

Disease	Biologic Drug	Age (Years)	Approval	References
CRSwNP	Omalizumab (anti-IgE)	/	No	[12,13,16]
	Dupilumab (anti-IL-4Rα)	≥18	Yes (EMA, FDA)	[12,13]
	Mepolizumab (anti-IL-5)	/	No	[12,13,15]
	Benralizumab (anti-IL-5Rα)	/	No	[12,13,14]
EoE	Dupilumab	≥12	Yes (FDA)	[27,28,29]
		1–11 ongoing	No	
	Cendakimab (anti-IL-13)	/	No	[30]
Anaphylaxis	Omalizumab	/	No	[33,34,35,36,37]
Food allergy	Omalizumab	/	No	[44,45,46,47,48]
	Dupilumab	/	No	[49,50]

## Data Availability

Not applicable.

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
