# Peer review of "New Indications of Biological Drugs in Allergic and Immunological Disorders: Beyond Asthma, Urticaria, and Atopic Dermatitis"

_biomedicines, 2023, doi:10.3390/biomedicines11020236_

Round 1
Reviewer 1 Report
1.Psoriasis is not an allergic disease, and away from the concept of this review regarding allergic diseases (all other diseases in this review are allergy-related), thus should not be discussed in this review and should be deleted.
2.Table 1
AIT= allergen immunotherapy is not the name of disease, so it is not correct to put this term on the present position in this table.
Where (in which countries) the Approval is obtained should be described for each therapy. Europe? USA?
Author Response
We thank the reviewer for his precious advise. As suggested, we have deleted psoriasis from the text by focusing exclusively on allergic pathologies. We also modified the table as requested.
Reviewer 2 Report
The manuscript contains interesting information regarding the biological treatment of various clinical manifestations of allergic diseases. Authors analyzed the available studies from the last years (mainly up to 10 years ago). The analysis is clearly divided by test units. In my opinion, this arrangement of the text increases the clarity of the study. The analysis is performed correctly. I believe that the manuscript can be published in its present form. However, I advise you to re-analyze the text in terms of linguistic correctness in order to remove typical minor stylistic and typographical errors. I have no other comments.
Author Response
We really thank the reviewer for his comments and his precious suggestions. We have re-analyzed the text and removed stylistic and typographical errors as requested.